# Experience with Photodynamic Therapy Using Indocyanine Green Liposomes for Refractory Cancer

**DOI:** 10.3390/jpm12071039

**Published:** 2022-06-24

**Authors:** Kensho Yorozu, Masaki Kaibori, Shintarou Kimura, Misa Ichikawa, Kosuke Matsui, Soichiro Kaneshige, Masanori Kobayashi, Daiki Jimbo, Yusuke Torikai, Yoshitaka Fukuzawa, Yoshiharu Okamoto

**Affiliations:** 1Yorozu Clinic, 1-118-4 Mihagino, Tottori 689-0202, Japan; verdy.jacky.nooki.trip@gmail.com; 2Department of Surgery, Kansai Medical University, 2-5-1 Shinmachi, Hirakata, Osaka 573-1191, Japan; kaibori@hirakata.kmu.ac.jp (M.K.); matsuik@hirakata.kmu.ac.jp (K.M.); 3StateArt Inc., StateArt Bild 2-9-12 Nihonbashi Horidomecho, Chuo-ku, Tokyo 103-0012, Japan; s.kimura@stateart.co.jp (S.K.); m.ichikawa@stateart.co.jp (M.I.); 4Department of Radiation Oncology, Okayama Central Hospital, 6-3 Ishimakitamachi, Okayama 700-0017, Japan; so9330@yahoo.co.jp; 5Serenclinic-Nagoya, 2F, Hisaya Park Bld, 4-14-2 Sakae, Naka-ku, Nagoya, Aichi 460-0008, Japan; kobyashi.masa@gmail.com; 6Aichi Medical University Hospital Preemptive & Integrative Medicine Center 1-1, Iwasaku Karimata, Nagakute, Aichi 480-1195, Japan; jimbo@jobs.jpn.org (D.J.); yofuku@aichi-med-u.ac.jp (Y.F.); 7Department of Veterinary Clinical Medicine, School of Veterinary Medicine, Tottori University, Tottori 680-8553, Japan; yokamoto@tottori-u.ac.jp

**Keywords:** photodynamic therapy, indocyanine green liposomes, multi-laser delivery system, Lentinula edodes mycelia, hydrogen gas, esophagus carcinoma, hypopharyngeal cancer

## Abstract

We reported the development of an effective cancer treatment using a multidisciplinary treatment, including photodynamic therapy (PDT) with indocyanine green (ICG) liposomes and a combination of Lentinula edodes mycelia (LEM) and hydrogen gas inhalation therapy. ICG liposomes were prepared by adding 5 mg of ICG to 50 mL liposomes. Later, 25 mL of ICG liposomes were diluted with 250 mL of 5% glucose solution and administered intravenously to the patient. We selected the multi-laser delivery system (MLDS), a laser irradiator for performing PDT. Further, the patients received a combination of LEM and hydrogen gas inhalation therapy throughout the treatment. We reported two cases of PDT therapy, one with middle intrathoracic esophagus carcinoma and the other with hypopharyngeal cancer. In the first case, the MLDS laser was directly attached to the endoscope and directed to the cancer area with wavelengths of 810 nm. After the treatment, a biopsy demonstrated no tumor recurrence. In the second case, the patient was treated with endovascular PDT using ICG liposomes and MLDS fiber optics. Later, tumor shrinkage was demonstrated after the first round and disappeared after six months. In conclusion, the present findings suggest that the effect of PDT using ICG liposomes with LEM and hydrogen gas may eradicate cancer without burdening patients by enhancing tumor immunity.

## 1. Introduction

Photodynamic therapy (PDT) irradiates cancer tissue directly with light in the same way as radiation therapy [1,2,3]. It is an extremely safe method as the laser itself does not show toxicity upon exposure to light; thus, only a drug designed to accumulate specifically in cancer cells will show toxicity [4]. Therefore, we focused on PDT using polymer drugs as a therapeutic method that efficiently elicits immunogenic cell death (ICD) in cancer cells with few side effects. Polymer drug conjugates are polymer-based prodrugs designed to increase the aqueous solubility of anticancer drugs, thereby enhancing bioavailability [5]. Photosensitizers used for PDT, especially porfimer sodium and talaporfin sodium, have been widely used in the medical field. However, these agents demonstrate serious drawbacks, such as slow excretion and reaction to visible light, forcing the patient to shade for several days.

In contrast, indocyanine green (ICG) is a photosensitizer that is sensitive to a wide wavelength, ranging from red to near-infrared (NIR), with peak absorption in the NIR region [2]. It is also well excreted, generates singlet oxygen, and generally demonstrates the same effect as porfimer sodium and talaporfin sodium. Singlet oxygen rapidly reacts with intracellular macromolecules, such as nucleic acids, proteins, and lipids, to injure cancer cells [6]. Cell injury caused by singlet oxygen causes endoplasmic reticulum stress, eliciting ICD and releasing damage-associated molecular patterns, such as calreticulin and high-mobility group box-1 [7,8]. In addition, ICG is expected to demonstrate an antitumor effect via photothermal therapy (PTT) because it generates heat by NIR irradiation [2]. In contrast, ICG is rapidly metabolized after intravenous administration, making it difficult to accumulate in cancerous tissues [9]. Recently, some studies have reported the use of drugs in which ICG has been converted into liposomes or micelles [10,11,12], which enables ICG to accumulate in cancer through the enhanced permeability and retention (EPR) effect [12].

Regarding CTLA-4 inhibitors, ipilimumab is considered mainstream among the antibody drugs [13,14] although Lentinula edodes mycelia (LEM) extracts may demonstrate the same effect [15]. LEM extract is a brown powder obtained by inoculating and culturing shiitake mycelium in a solid medium composed of bagus and rice bran, extracting LEM with hot water, and then drying and grinding the extract to a powder [15]. Researchers have reported that LEM improves the immunosuppressive state centered on increasing Tregs in cancer-bearing mice and delays tumor growth by restoring antitumor immunity [16]. LEM also contains multiple components with an antioxidant and antitumor effect rather than a single substance [17].

In contrast, hydrogen gas reduces inflammation of the body by removing reactive oxygen species such as singlet oxygen, superoxide, hydrogen peroxide, and hydroxyl radical [18]. Akagi et al. reported that the inhalation of hydrogen gas reduced PD-1 expression in positive cytotoxic T-lymphocyte (CTLs) and increased negative CTLs [19].

In this study, we reported the development of an effective cancer treatment using a multidisciplinary treatment based on PDT using ICG liposomes and a combination of LEM and hydrogen gas inhalation therapy.

## 2. Materials and Methods

### 2.1. Preparation of ICG Liposomes and PDT

1,2-Dimyristoyl-sn-glycero-3-phosphocholine (DMPC) was purchased from Yuka-Sangyo Co., Ltd. (Tokyo, Japan), and Diagnogreen (ICG) was purchased from Daiichi Sankyo Co., Ltd. (Tokyo, Japan). We dissolved DMPC in a 5% glucose solution to a concentration of 8.85 mM prepared by 40 kHz ultrasonic irradiation at 45 °C for 60 min using Bransonic^®^ CPX8800H-J Ultrasonic Cleaner (Branson Ultrasonic Co., Ltd., Danbury, CT, USA). Then, it was also filtered and sterilized with a 0.20-mm pore size filter to prepare liposomes. The size of liposomes was determined to be approximately 100 nm using ELSZ-2000 (Otsuka Electronics Co., Ltd., Osaka, Japan). The liposomalization of ICG was determined via gel filtration chromatography using the PD-10 Sephadex G-25 gel filtration column (Cytiva, MA, USA).

A total of 25 mL of ICG liposomes was diluted with 250 mL of 5% glucose solution and administered intravenously to the patients (Figure 1A). We selected a globally approved multi-laser delivery system (MLDS) (Weber, Germany) as a laser irradiator for performing PDT. MLDS can output wavelengths of 375–870 nm and irradiate up to 6 wavelengths at 12 locations simultaneously (Figure 1B). In addition, MLDS is suitable for low-level laser therapy to be safely inserted into blood vessels and lymph vessels. We uniquely modified the device and attached it to the endoscope (Olympus LUCERA 260 GIF-XP260N, Tokyo, Japan), as shown in Figure 1C. Four fibers were used, two at 635 nm and two at 810 nm. Light was irradiated at 500 mW/cm^2^ per fiber for 40 min. The total amount of energy was 4800 J/cm^2^ (energy of two 635 nm fibers: 2400 J/cm^2^, two 810 nm fibers: 2400 J/cm^2^). Patients provided informed consent before PDT using ICG liposomes. The institutional ethics committee of the Japanese Organization for Safety Assessment of Clinical Research approved the study design (reference number: Sacrj-20201028-07).

### 2.2. Combination of LEM and Hydrogen Gas Therapies

Shiitagen Pro (LEM) was purchased from Kobayashi Pharmaceutical Co., Ltd. (Osakam, Japan). Patients received an oral dose of 1200–1800 mg of LEM twice or three times a day for the PDT period. The Hycellvator PF72 (Helix Co., Ltd., Tokyo, Japan) hydrogen gas inhaler was installed in the patients’ homes. Patients inhaled hydrogen gas at 866 mL/min daily for 1–3 h via Hycellvator PF72 for the PDT period. Patients provided informed consent before these therapies were started.

## 3. Results

### 3.1. Case of PDT Combined with Radiotherapy in Middle Intrathoracic Esophagus Carcinoma

A Japanese man in his 60s who had a feeling of discomfort in the chest was diagnosed with stage II (T1bN0M0) esophagus carcinoma via upper endoscopy at another hospital in Tottori prefecture, Japan, in July 2019. Esophageal cancer was determined by pathological findings. The general hospital recommended surgery, but the patient chose to undergo conservative treatment at our hospital. In September 2019, endoscopy showed a torose lesion (2.5 × 1.0 cm) in the upper intrathoracic esophagus (Figure 2A). The patient was treated with 2-Gy radiation per session for a total of 33 sessions; thus, 66-Gy was irradiated over 3 months. In November 2019, tumor shrinkage was observed (2.0 × 0.5 cm) (Figure 2B). Later, in late November 2019, 50 mL of ICG liposomes was intravenously administered to the patient combined with endoscopic PDT in approximately 1 h. The MLDS laser device was directly attached to the endoscope with a wavelength of 810 nm targeting the cancer area (Figure 2C). PDT was performed using MLDS laser once for 20 min. Between December 2019 and March 2020, the patient received six PDT sessions. In December 2020, endoscopy showed tumor disappearance, and biopsy showed no recurrence by pathological findings (Figure 2D,E). Computed tomography (CT) images of esophageal cancer before and after PDT are shown in Figure 2F,G. Compared with before PDT treatment (Figure 2F), there was a decrease in mucosal surface staining after treatment (Figure 2G). In addition, a slight decrease in wall thickening was seen in the direction of the left front wall (indicated by the arrow). This patient underwent a combination of LEM and hydrogen gas inhalation therapy throughout the treatment. No adverse events were observed with these treatments.

### 3.2. Case of the Effect of PDT on Hypopharyngeal Cancer

In February 2018, a 40-year-old woman with left cervical lymphadenopathy was diagnosed with hypopharyngeal cancer at another hospital in Okayama prefecture, Japan. Hypopharyngeal cancer was determined by pathological findings via needle biopsy. She tried chemotherapy with some anticancer drugs (cisplatin, 5-fluorouracil, and docetaxel hydrate) and irradiation therapy, which were ineffective; therefore, she was referred to our hospital in December 2019. The size of the tumor was 16.5 × 15.3 cm. We planned photoimmunotherapy with ICG liposomes combined with Moths paste [20] via orthopedic surgery for self-destructive tissue. In January 2020, 50 mL of ICG liposomes was infused via intravenous drip in approximately 1 h. The next day, the neck was irradiated with an 810-nm laser for 40 min using an MLDS optical fiber (Figure 3A). Tumor shrinkage was observed 1 month after the first treatment (14.0 × 9.5 cm) (Figure 3B). Later, PDT using ICG liposomes was performed twice in February (Figure 3C). In May 2020, the skin ulcers caused by the tumors were significantly repaired (Figure 3D). CT images of hypopharyngeal cancer before and during PDT are shown in Figure 3E,F. These figures show improvement of lymphadenopathy with PDT treatment (inside the circle on the right side), and a clear reduction in tumor size was observed (inside the circle on the left side). The patient received three PDT sessions in addition to a combination of LEM and hydrogen gas inhalation therapy throughout the treatment. No adverse events were observed with these treatments.

## 4. Discussion

This study demonstrated two cases involving LEM and hydrogen gas in combination with PDT in the treatment plan. So far, to the best of our knowledge, the effects of liposomal or micellized ICG have been reported only in animal experiments [10,11,12]. Thus, this report is the first to apply the treatment to humans. Advanced cancers require multifaceted treatments. Ideally, it is important to induce ICD in cancer tissues without damaging the immune system and present antigens in antigen-presenting cells (APCs) [21] in addition to approaching the immune checkpoint with substances that reduce side effects [22]. Further, individuals receiving PDT using ICG liposomes can undergo PTT simultaneously. We hypothesized that these treatments induce tumor tissue-specific ICD and activate CTLs using hydrogen gas and LEM (Figure 4). In our study, patients with advanced esophageal and hypopharyngeal cancer were treated with LEM and hydrogen gas for immune checkpoint inhibition, while undergoing PDT using ICG liposomes. PDT using ICG liposomes is not cancer-specific, but it is expected to target cancer through the EPR effect selectively [23]. Our group recently reported that EPR enhancers improve the efficiency of the EPR effect of macromolecular drugs [24]. Further enhancing the accumulation of ICG liposomes in cancer is possible by improving the EPR effect and incorporating polyethylene glycosylated hyaluronic acid or similar agents that bind to a cancer-specific receptor in the liposome [25].

Previous studies have shown that LEM suppresses B16 melanoma growth by alleviating regulatory T cell–mediated immunosuppression [16]. In contrast, Akagi et al. reported that hydrogen gas inhalation therapy enhances the effect of nivolumab by acting on the mitochondria of exhausted CD8 positive T cells, which is a poor prognosis factor for cancer, and converting them into active CD8 positive T cells [19]. They suggested that hydrogen gas acts on the PD1/PDL1 pathway, allowing CTLs to efficiently attack cancer cells. We believe that these two different immune checkpoint suppressive effects led to tumor eradication of PDT using ICG in these two cases.

In conclusion, PDT using ICG liposomes is an inexpensive and simple treatment method. Used in combination with LEM and hydrogen gas, that demonstrate no side effects, this treatment is expected to eradicate cancer without burdening patients, by enhancing the body’s immunity against tumors.

## Figures and Tables

**Figure 1 jpm-12-01039-f001:**
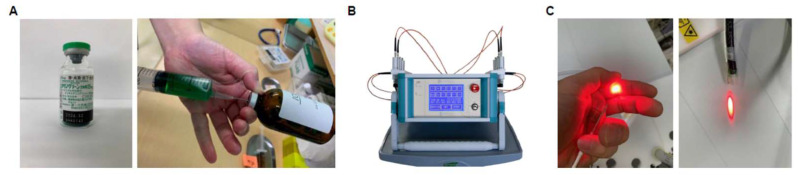
(**A**) Indocyanine green (ICG) (**left**). Five mg of ICG was dissolved in water and mixed with an 8.85-mM liposome solution as ICG liposomes (**right**). (**B**) Image of multi-laser delivery system (MLDS). (**C**) MLDS optical fiber shining a 635-nm laser (**left**). The device was modified and installed so that the laser could be applied directly to the target region endoscopically (**right**).

**Figure 2 jpm-12-01039-f002:**
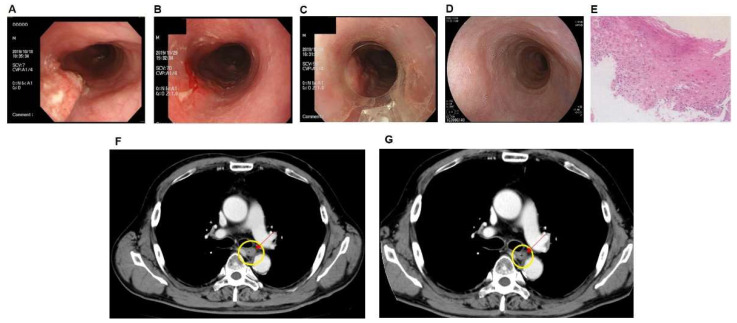
(**A**) Endoscopic images of middle intrathoracic esophagus carcinoma in the patient before treatment. An irregular elevation was found in the upper intrathoracic esophagus. (**B**) After radiation therapy, endoscopic images around the tumor lesion (66 Gy/33 times). (**C**) Endoscopic images after photodynamic therapy using ICG liposomes. The 635-nm and 810-nm lasers were applied to the periphery of the lesion via optical fiber for 20 min. (**D**) Endoscopic images after 1 year of treatment completion. Endoscopic disappearance was observed, and no recurrence was observed after over 1 year. (**E**) One year after the completion of treatment, the stratified squamous epithelium of the esophagus was thickened with parakeratosis in pathological section observation. No atypia was found; thus, no malignant finding was found in this biopsy. (**F**) CT image before photodynamic therapy. (**G**) CT image after photodynamic therapy.

**Figure 3 jpm-12-01039-f003:**
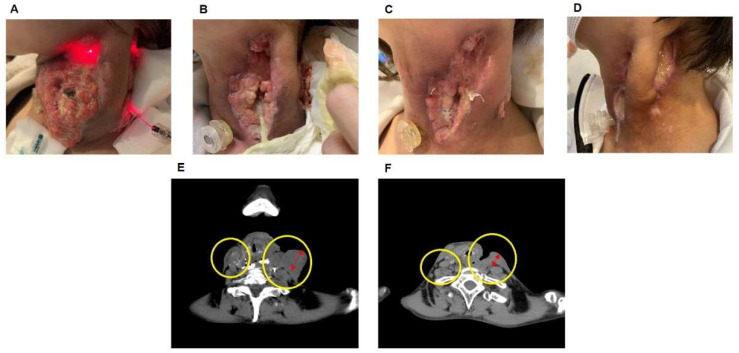
(**A**) Image immediately after the start of photodynamic therapy (PDT) using ICG liposomes. Laser irradiation of 635 nm and 810 nm was applied to the neck via optical fiber for 40 min. (**B**,**C**) A similar PDT with ICG liposomes was performed three additional times every other month. (**D**) Four months after the start of treatment, no tumor tissue was observed. (**E**) CT image before photodynamic therapy. (**F**) CT image during photodynamic therapy.

**Figure 4 jpm-12-01039-f004:**
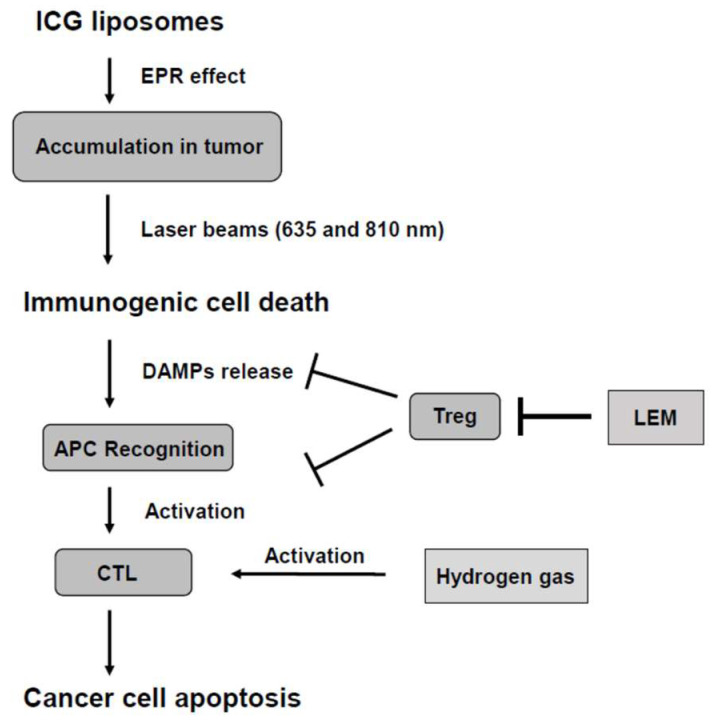
Mechanism of PDT using ICG liposomes. ICG; indocyanine green, EPR; enhanced permeability and retention, DAMPs; damage-associated molecular patterns, Treg; regulatory T-lymphocyte, LEM; Lentinura edodes mycelium, APC; antigen-presenting cell, CTL; cytotoxic T-lymphocyte.

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
