# Peer review of "Experience with Photodynamic Therapy Using Indocyanine Green Liposomes for Refractory Cancer"

_jpm, 2022, doi:10.3390/jpm12071039_

Round 1

Reviewer 1 Report

The manuscript described two case reports of treating refractory cancer by using a combination of indocyanine green (ICG) liposomes mediated photodynamic therapy (PDT), Lentinula edodes mycelia (LEM) extract and hydrogen gas inhalation therapy. The introduction provides sufficient background. However, the treatment modality was different between two patients, including PDT sessions, and period of irradiation time. It seemed to be a trial-and-error approach. The results of both cases showed tumor disappearance based on the images of tumor. More date such as CT or biomarkers should be provided to demonstrate no tumor recurrence. In addition, references must be numbered consecutively in the order they are first mentioned.

Author Response

Major points

  1. The manuscript described two case reports of treating refractory cancer by using a combination of indocyanine green (ICG) liposomes mediated photodynamic therapy (PDT), Lentinula edodes mycelia (LEM) extract and hydrogen gas inhalation therapy. The introduction provides sufficient background. However, the treatment modality was different between two patients, including PDT sessions, and period of irradiation time. It seemed to be a trial-and-error approach. The results of both cases showed tumor disappearance based on the images of tumor. More date such as CT or biomarkers should be provided to demonstrate no tumor recurrence. In addition, references must be numbered consecutively in the order they are first mentioned.

Response

We appreciate these comments. According to your suggestion, we added CT images of patients before and after PDT treatment to Figures 2 and 3 (Cases 1 and 2, respectively), and added the following text:

RESULTS (lines 139–143):

Computed tomography (CT) images of esophageal cancer before and after PDT are shown in Fig. 2F,G. Compared with before PDT treatment (Fig. 2F), there was a decrease in mucosal surface staining after treatment (Fig. 2G). In addition, a slight decrease in wall thickening was seen in the direction of the left front wall (indicated by the arrow).

(lines 170–174):

In May 2020, the skin ulcers caused by the tumors were significantly repaired (Fig. 3D). CT images of hypopharyngeal cancer before and during PDT are shown in Fig. 3E,F. These figures show improvement of lymphadenopathy with PDT treatment (inside the circle on the right side), and a clear reduction in tumor size was observed (inside the circle on the left side).

Reviewer 2 Report

The authors  demonstrated  a   multimodal treatment of esophagus and  hypopharyngeal cancer  involving  photodynamic therapy using indocyanine green liposomes,  oral intake of   Lentinula edodes mycelia  and hydrogen gas inhalation therapy. The report is clearly presented with adequate discussion. It will be further improved if  the mechanism proposed in Fig. 4 is also investigated. 

Author Response

Thank you for your valuable comments. We would like to further elucidate the mechanism of action in Fig. 4 by accumulating a greater number of cases.

Reviewer 3 Report

Abstract

   Brief information about light parameters e.g. power density (mW/cm2), and energy density (J/cm2) is needed.

   Parameters regarding PDT must be disclosed

  . 

Author Response

Major points

Brief information about light parameters e.g. power density (mW/cm2), and energy density (J/cm2) is needed. Parameters regarding PDT must be disclosed

Response

Thank you for your valuable comments.

According to your suggestion, we added the following text:

Materials and Methods (lines 104–106):

Four fibers were used, two at 635 nm and two at 810 nm. Light was irradiated at 500 mW/cm2 per fiber for 40 minutes. The total amount of energy was 4800 J/cm2 (energy of two 635 nm fibers: 2400 J/cm2, two 810 nm fibers: 2400 J/cm2).

Round 2

Reviewer 1 Report

The manuscript has been revised according to the reviewers' suggestion, which may be accepted in its present form.